

**Construction of surface air temperature series of Qingdao in China for the period 1899 to 2014**
Yan Li[1], Birger Tinz[2], Hans von Storch[3], Qingyuan Wang[4], Qingliang Zhou[5]
[1]National Marine Data and Information Service, Tianjin, People's Republic of China
[2]German Meteorological Service, Hamburg, Germany
[3]Helmholtz-Zentrum Geesthacht, Centre for Materials and Coastal Research, Hamburg, Germany
[4]Tianjin Meteorological Observatory, Tianjin, People's Republic of China
[5]National Meteorological Center, China Meteorological Administration, Beijing, People's Republic of China
*Correspondence to*: Yan Li (ly_nmdis@163.com)
**Abstract.**   We present a homogenized time series surface air temperature at 2 meters (SAT) for the
city of Qingdao in China from 1899 to 2014. This series is derived from three data sources: newly
digitized and homogenized observations of German National Meteorological Service from 1899 to 1913;
National observation data of China Meteorological Administration (CMA) from 1961-2014 and a
gridded data set of Willmott and Matsuura in Delaware to fill the gap from 1914 to 1959. Based on this
new series, long-term trends are described. The SAT in Qingdao has a significant warming trend of
0.11 ℃ (10 yr)$^{-1}$ during 1899-2014. The coldest period occurred in 1909-1918 and the warmest period
occurred in 1999-2008. For the seasonal mean SAT, the most significant warming can be found in spring,
followed by winter.  Access to the data is provided in excel and archived by Deutscher Wetterdienst
(DWD)   web   page   under   overseas   stations   of   the   Deutsche   Seewarte
(http://www.dwd.de/EN/ourservices/overseas_stations/ueberseedoku/doi_qingdao.html)  or  be  freely
available at https://dx.doi.org/10.5676/DWD/Qing_v1.



## 1 Introduction

Surface air temperature at 2 meter (hereinafter referred to as SAT) is one of the most important climate elements influencing the biosphere and human activities. Systematical observations in China on national scale started in 1951. However, the 60 years length of the SAT dataset seems insufficiently long to understand the long-term trend and interdecadal variability. For detecting changes beyond the range of natural variations (e.g., Zorita et al., 2008) and for attributing such a change to plausible driver (a concept introduced by Hasselmann (1979) known as "detection and attribution) longer observational series are needed. Therefore, the changes of temperatures in China in the past more than 100 years need to be investigated in more details (Qian and Zhu et al., 2001; Soon, et al., 2011).

Several annual-mean SAT series for China commencing in the late 19[th] century have been constructed (Lin et al.,1995; Mitchell et al., 2002; Wang et al., 2004; Tang and Ren, 2005; Tang et al., 2009; Soon, et al., 2011; Compo et al., 2011). Using the reconstructions of mean temperature for 10-regions and estimates from historical documentary records, from ice core records, and from tree ring records one such series was presented by Wang and Gong (2000), Wang et al. (2004), and Lin et al. (1995). Others determined monthly average values of the daily maximum and minimum SAT and derived from these monthly mean SAT estimate (Tang and Ren, 2005; Tang et al., 2009). However, these studies exhibit widely different linear trends of nationally-averaged SAT namely 0.3 ℃/100 years and 1.11 ℃/100 years. These differences stem mostly from differences in the period before 1951, where there is greater uncertainty in the time series (Tang et al., 2009; Soon et al., 2011). This phenomenon is caused by the sparse temporal and spatial resolution in the earlier time in some degree.

Fortunately, the International Atmospheric Circulation Reconstructions over the Earth (ACRE) project was set up in 2008. One aim of ACRE is to link international meteorological organizations for the recovery, quality control and consolidation of global terrestrial and marine instrumental surface data



of the last 250 years (Allan et al., 2011, 2016). Among others, also the German Meteorological Service
(Deutscher Wetterdienst, DWD) in Hamburg supports this project with huge archives of historical
handwritten journals of weather observation. Archived data from about 1,500 *Overseas stations* of the
Deutsche Seewarte (German Marine Observatory, "Deutsche Seewarte", Hamburg, existing from 1875
to 1945) in the late 19th and the first 20th century have been digitized and quality controlled (Kaspar et
al., 2015, see also https://www.dwd.dw/EN/ourservices/overseas_stations/ueberseestationen.html).
Most of the stations existed in the former German colonies in Africa, islands in the tropical Pacific as
well as the "Kiautschou Bay" around Qingdao (German name: "Tsingtau") (see Figure 1a). The
Kiautschou Bay territory was leased by the German government from the Chinese Qing  dynasty
(https://www.bundesarchiv.de/oeffentlichkeitsarbeit/bilder_dokumente/00765/index.html.de,        in
German). It was established in 1898 and ended in 1914 with the beginning World War I.   The digitized
data and the original documents of the Qingdao Station (Figure 1b) for the period 1899-1913 were
handed over to China Meteorological Administration (CMA) and Municipal Weather Service of
Qingdao in 2008, 2014 and 2015. We use these data for describing temperature variations in 1899-1913.
Here, there are two questions need to be considered: 1) How can we make a good use of these data;
2) What can be obtained from these data in climate change studies. This study attempts to use the
Qingdao station as an example to objectively establish a new homogenized monthly mean SAT series
back to the 1899. Then, the derived time series are used to analyze the characteristics of climate
variability in Qingdao where rapid industrial developments have taken place.



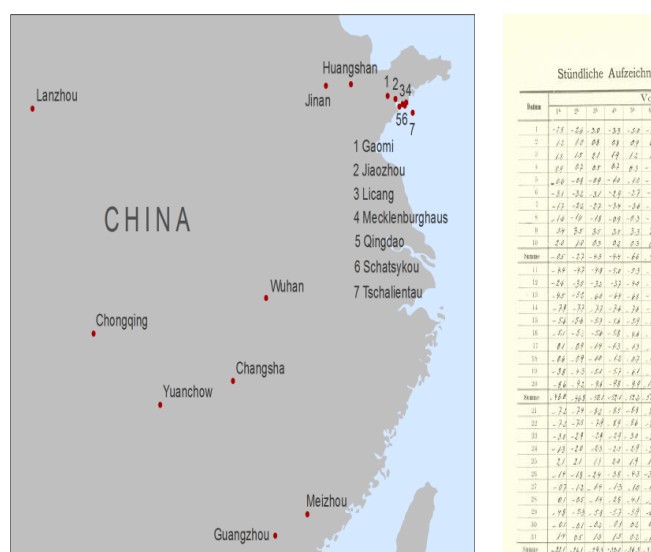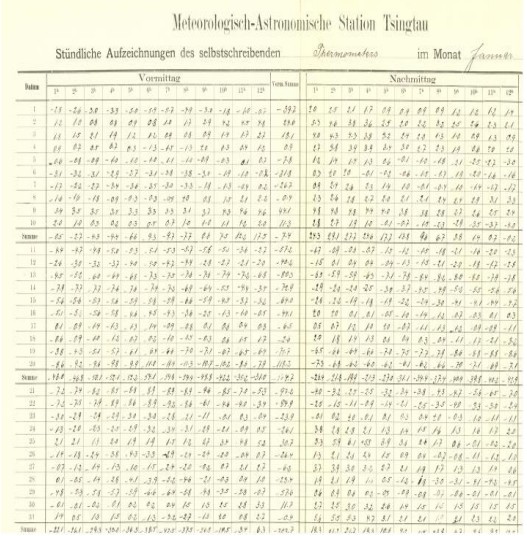

**Figure 1**. Positions of the 14 stations of the German Marine Observatory in China (left); Handwritten observations of Qingdao (right) from archive German Meteorolgical Service in Hamburg, Germany.

## 2. Data and methods

### 2.1 Data Sources

Several data sources are used in the study. The observed sub-daily SAT records and the associated metadata of Qingdao (1899 to 1913) (Table 1) have been archived by DWD. Note that these SAT records from 1905 to 1914 have high temporal resolution of 24 records of each day. The monthly and homogenized SAT of Qingdao Station from 1961 to 2013 is selected from CMA which has developed the first national homogenized temperature data set (Li and Yan, 2009) and its updated version (Xu et al., 2013). Moreover, three gridded SAT data sets starting from the late 19[th] have been used in the construction: 1) the monthly mean SAT of the global precipitation and temperature of Willmott and Matsuura which is developed in the Department of Geography, University of Delaware referred as SAT W&M v4.01 (Willmott and Matsuura, 2012); 2) the monthly mean SAT data of the Climatic Research Unit (Harris et al., 2013) , referred as CRU TS3.230; 3) and the monthly mean SAT data of the 20[th]





Century Reanalysis version 2c, referred as 20CR v2c (Compo, et al., 2011). More details about the three
datasets are shown in Table 2.

**Table 1**. Coordinates, height and daily observation times of Qingdao and Schatsykou in the earlier time.

| Station | Latitude/Longitude | Height | Time period | Daily observation times |
|---|---|---|---|---|
| Qingdao | 36 ̊04'N/120 ̊17'E | 24m | 01.01.1899 - 30.06.1899 | 8am, 2pm, 8pm |
| Qingdao | 36 ̊04'N/120 ̊17'E | 24m | 01.07.1899 - 30.04.1905 | 7am, 2pm, 9pm |
| Qingdao | 36 ̊04'N/120 ̊19'E | 50m | 01.05.1905 - 31.08.1905 | 7am, 2pm, 9pm |
| Qingdao | 36 ̊04'N/120 ̊19'E | 78.6m | 01.09.1905 - 31.03.1914 | Hourly (24 times a day) |
| Schatsykou | 36 ̊06'N/120 ̊32'E | 20m | 01.12.1898 - 30.04.1899 | 8am, 2pm, 8pm |
| Schatsykou | 36 ̊06'N/120 ̊32'E | 20m | 01.07.1900 - 31.10.1901  01.05.1903 - 30.09.1909 | 7am, 2pm, 8pm |


**Table 2**. Three gridded SAT data sets that are used in this study

| Name and Web address | Period | Temporal resolution | Spatial resolution |
|---|---|---|---|
| SAT W&M v4.01  http:/esrl.noaa.gov/psd/data/gridded/data.UDel_ AirT_Precip.html | 1900–2014 | monthly | 1 ̊x 1 ̊ |
| CRU TS3.230  https://crudata.uea.ac.uk/cru/data/hrg/cru_ts_3.23 | 1901–2014 | monthly | 0.5 ̊x 0.5 ̊ |
| 20CR v2c  http://rda.ucar.edu/datasets/ds131.2/ | 1851-2014 | monthly | 2 ̊x 2 ̊ |




## 2.2 Testing for significance of correlations and trends


When test the significance of correlation between two time series and the significance of the presence
of trends in a time series, an important question needs to considering, that is, how large sample
correlations and sample trends could be, even if the stochastic processes, which generate the series, are
not correlated at all and exhibit no trends. Firstly, we have to make an assumption, namely the
processes X and Y share no correlation, or segments of length L of the process have no trend.
Standard procedures are available in the literature, namely p value for correlations and Mann-Kendall
for trends (e.g., Kulkarni and von Storch 1995; von Storch and Zwiers 1999) that there are "no
correlations" between the underlying processes and trends can hardly appear in limited segments of an
infinite stationary time series.
In the case of correlations, the assumption is that the underlying processes are stationary (free of
systematic trends) and serially independent, i.e., $X_t$ and $X_{t+1}$ for any t are independent. In the case
of trends, the assumption is the independence of $X_t'$. However, in geophysical cases, these assumptions
are not satisfied- the result is that the null hypotheses are often falsely rejected (i.e., in cases where there
are no correlations or no trends (cf. Kulkarni and von Storch 1995)) than stipulated by the significance
level (normally 5%).
A practical remedy for avoiding such errors is to deal with normalized series (mean =0, standard
deviation=1) $X_t'$ (and $Y_t'$) as follows:
(1) "detrend" the time series before testing for correlations between two time series $X_t$ and $Y_t$.
Firstly, determining the linear fit $f_t^X$ and $f_t^Y$, and then do the hypothesis testing with $X_t' = X_t -$
$f_t^X$ and $Y_t' = Y_t - f_t^Y$



(2) "prewhiten" the time series, by first determining the sample autocorrelation $\alpha = 1/L \sum_t X_t X_{t+1}$
of the time series $X_t$ of length L, and forming a series $X_t' = X_t - \alpha X_t$, and then testing for the
null hypothesis of no trend.
To both cases, the standard routines are applied. If the null hypothesis is rejected at the stipulated
significance level of 5%, then the sample trend $f_t^X$, or the sample correlation $1/L \sum_t X_t X_{t+1}$, is
"significant".
In this paper, four seasonal mean SAT is defined by calculating the average of each three-month:
December (-1yr)-January-February (Winter), March-April-May (Spring), June-July- August (Summer)
and September-October-November (Autumn). Then, the linear trend of each season is also shown in
this study; also the significance of each trend has been test.

**3. Long term evaluation of the SAT in Qingdao**
**3.1 Processing of the earlier data from 1899 to 1913**
**3.1.1 Quality control and construction of missing data**
The earlier SAT data from 1899 to 1913 have been digitized manually and passed through a
quality check. The quality checking routine of DWD starts with a formal check, followed by
climatological, temporal, repetition and a consistency checks (Leiding, et al., 2016). From April to
December 1901, the original observations of Qingdao are not available.  We filled in estimates of these
missing values using the SAT time series of a neighboring station. We find that the value and variability
of SAT monthly time series from 1900 July to 1901 December in Schatsykou exhibits a good agreement
with these in Qingdao (Figure 2). Consequently, the SAT data in Schatsykou is merged into the SAT
data of Qingdao to fill with the missing data.





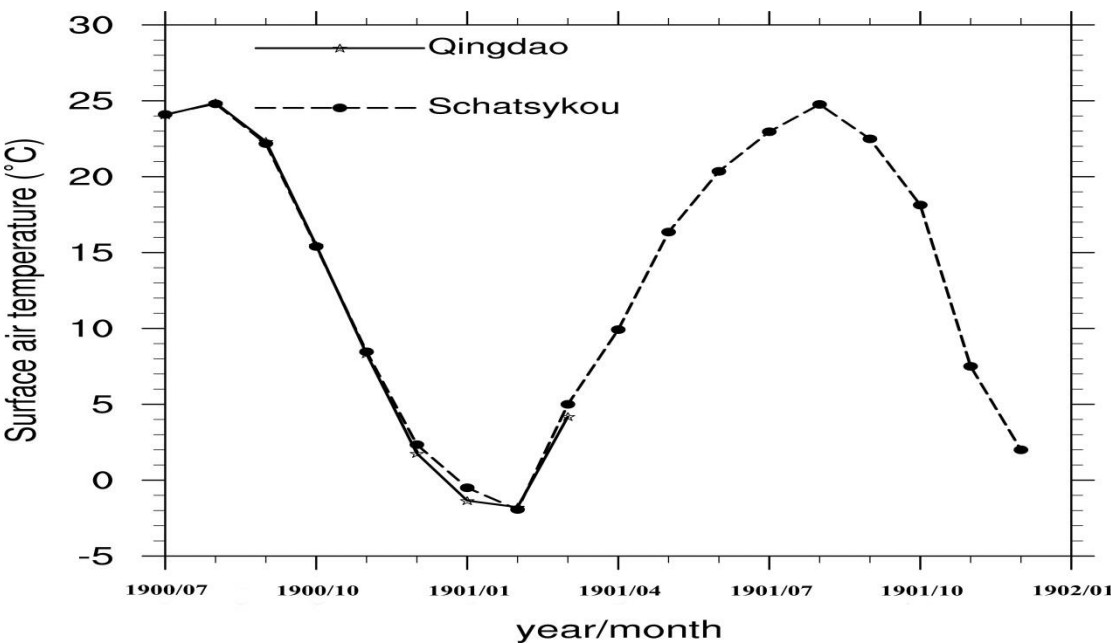


**Figure 2.**    Comparison of the monthly mean SAT in Qingdao (solid line) with that in Schatsykou (dashed line) during
July 1900 to December 1901.

Here, we have to point out that these quality checks above account for errors in coding and
archiving, but is not efficient in dealing with inhomogeneities (Karl et al. 1993). In fact, changes of
station height and changes of daily observational times (Table 1) can affect the SAT during 1899 to
1913. It is important in observational studies that the data used should to be homogeneous (Trewin,
2010; Wang et al., 2014; Li et al., 2016). Thus, the homogenization of the SAT time series from 1899 to
1913 needs to be done in the next step.

**3.1.2 Homogenization of SAT time series from 1899 to 1913**

Inhomogeneities in land-based observations of air temperature may dampen, or introduce noise to
estimate of long-term air temperature trends.    The SAT data from 1961 to 2014 have been



homogenized by CMA (Xu, et al., 2013). Here, we pay more attention on the detecting and adjusting
of the SAT homogeneity from 1899 to 1913 which is newly digitized without homogenization.
Details of non-climatic factors were recorded in the metadata back to Januray 1899 (Table 1).
Among these factors, the changes of observation height and daily observation times are the main causes
of inhomogeneities. The observation heights have been changed three times since 1899: 24 m (1[th]
January 1899-30[th] April 1905); 50 m (1[th] May 1905- 31[th] August 1905); 78.6 m (1[th] September 1905
-31[th] March 1914). The observing schedule was also changed in July 1899 from local time 08, 14, 20, to
07, 14, 21; and the observing schedule was changed again in April 1905 from local time 07, 14, 21 to
hourly (24 times a day). In different periods the daily mean was calculated by the different formulas:
$T_{daily}(I) = (T_{07} + T_{14} + 2 \times T_{21})/4$         (1)
$T_{daily}(II) = (T_{08} + T_{14} + 2 \times T_{20})/4$         (2)
$T_{daily}(III) = (T_{01} + T_{02} + \cdots + T_{24})/24$       (3)
In order to adjust the inhomogenities caused by observation height change and observation times
change, the best way is to find neighboring reference series and then modify the candidate series based
on several mathematical methods. But actually it is hard and even impossible to find a reference series
in such early times. In this case, air temperature in Qingdao was transformed into temperature at sea
level using an average environmental lapse rate (6.0 ℃/km). Furthermore, to avoid the inconsistency of
calculating a daily mean SAT from different observational times, the deviation caused by change of
observational times is assessed firstly. We use $T_{daily}(I)$, $T_{daily}(II)$, $T_{daily}(III)$ to calculate the daily mean
SAT, respectively using the observation data from 1908 to 1913 which were recorded 24 times a day.
Then, monthly means in the whole period (1908-1913) are obtained by the average of the days in each
month.





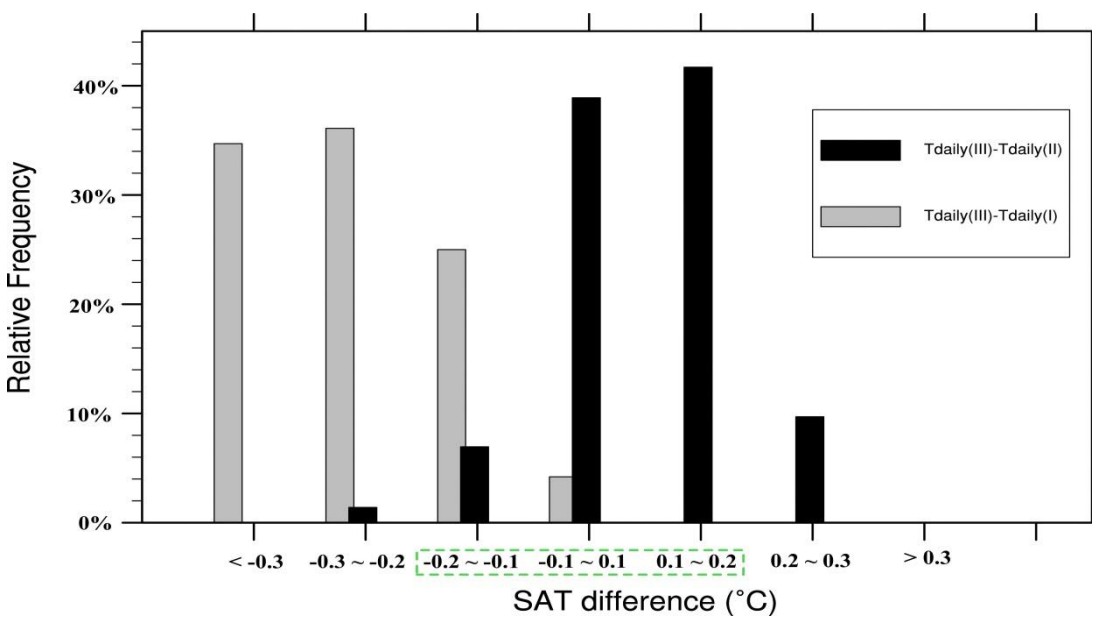


**Figure 3**. Distribution of difference between the daily mean temperature $T_{daily}(III)$ and $T_{daily}(I)$ (black bars) and
difference between $T_{daily}(III)$ and $T_{daily}(II)$ (grey bars) in the 72 months from January 1908 to December 1913 in
Qingdao. The SAT differences in green frame are considered as physically insignificant.

Generally, the SAT calculated by $T_{daily}(III)$ can represent the "real" daily mean SAT best. Thus, we
calculate the frequency distributions of deviation between the monthly mean SAT of $T_{daily}(III)$ and
$T_{daily}(II)$, as well as $T_{daily}(I)$ from January 1908 to December 1913 to assess the bias caused by different
calculation formulas. Results are shown in Figure 3. The majority of differences between $T_{daily}(III)$ and
$T_{daily}(II)$ are from -0.1 ℃ to 0.2 ℃. These discrepancies are small and can be ignored as physically
insignificant. So the SAT from July 1899 to August 1905 need not to adjust the bias caused by
$T_{daily}(II)$. However, the mean deviation between $T_{daily}(III)$ and $T_{daily}(I)$ is -0.27 ℃ and 35% of the
differences are larger than -0.3 ℃. It suggests the SAT calculated by $T_{daily}(I)$ has significant
time-varying biases with that calculated by $T_{daily}(III)$, about 0.27℃. Consequently, the SAT during
January 1899 to June 1899 has applied to an adjustment of -0.27 ℃. Figure 4 exhibits the annual mean
SAT series before and after adjustment.

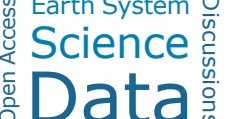



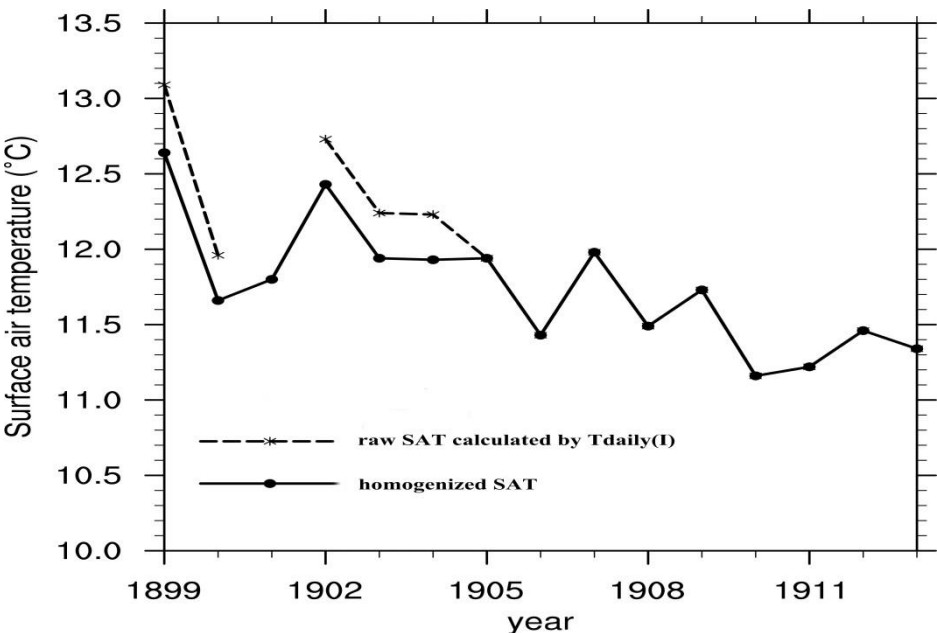

**Figure 4**. Annual mean SAT series (℃) of the raw and homogenized recorded at Qingdao from 1899 to 1913. Black line is the original observation series; red line is the homogenized series.

## 3.2 Construction of SAT series in the past 115 years (1899-2014)

Previous study point out that the observation SAT from 1914 to 1960 was discontinuous with large number of missing data (Cao et al., 2013).  To obtain continuous series of SAT during the past 116 years, three gridded SAT datasets have been investigated which were used to fill with the gap from 1914 to 1959. We calculated the correlation coefficient between the three SAT series with observational SAT from DWD and CMA in Qingdao station (hereinafter referred to as SAT OBS) in two periods (1899-1913 and 1960-2014) after detrending described in Section 2.2. The results show that all of the correlation coefficients are statistically significant on 95% confidence interval (Table 3). The highest correlation coefficients (r=0.98 and r=0.95) are found between SAT OBS and SAT W&M v4.01. However, the correlation with the estimate of the CRUTS3.230 shows rather low values, which are considerably smaller than those obtained with W&M v4.1 or 20CRv2c.



Then, we compare the three annual mean SAT time series with the SAT OBS in Figure 5. It can be
seen that except the 20CR v2c, the annual mean SAT series of W&M v4.01 and CRU TS 3.230 both
have the similar climate variability with that of the SAT OBS. Interestingly, the difference between
20CRv2c and CRUTS 3.230 shows marked non-stationarities. In the first years, both series are similar,
but exhibit relatively smaller interannual variability. Since about 1920 until 1950, the temperatures of
20CRv2c are mostly smaller than the CRUTS 3.320. But since about 1960 until 2005, the yearly means
of 20CRv2c are strongly larger than CRUTS3.320. The abrupt change in 20CRv2c around 1960 is not
replicated in the W&Mv4.01 series, and we suggest that this jump is an artifact in the analysis of
20CRv2c. Other non-stationarities have been found in the 20CR analyses (e.g., Krueger et al., 2013)
and we suggest to rely more on the other two descriptions of past temperature variations. However,
there is relatively large systematic difference between the SAT OBS and the CRU TS3.230 data and the
difference value even exceed 3.5 ℃ (see also Cao et al., 2013). Based on these findings, we use SAT
W&M v4.01 for filling the gap in the observational series between 1914 and 1959 to obtain continuous
SAT series in Qingdao.
Using linear regression method, each monthly mean SAT OBS in each period can be estimated
from SAT W&M v4.01. Take SAT OBS in January for example, SAT OBS = (SAT W&M v4.01 +
0.62)/0.95 during 1899 to 1913; SAT OBS = (SAT W&M v4.01 + 1.15/0.94) during 1960 to 2014.
Then, we give the estimated linear relationship during 1914 to 1959, SAT OBS = (SAT W&M v4.01 +
0.85)/0.95.   Consequently, the monthly mean SAT OBS in the period of 1914 to 1959 in January was
obtained.   The SAT series in Qingdao from 1899 to 2014 is shown in Figure 6.





**Table 3**. Correlation coefficients between the different international SAT time series and the observational SAT time
series in Qingdao.   All of these time series have been detrended. The largest correlation coefficients are in bold.

| Period | SAT OBS&20CRv2c | SAT OBS&SAT W&M v4.01 | SAT OBS&CRUTS3.230 |
|---|---|---|---|
| 1899-1913 | 0.47 | **0.98** | -0.27 |
| 1961-2013 | 0.54 | **0.95** | 0.92 |


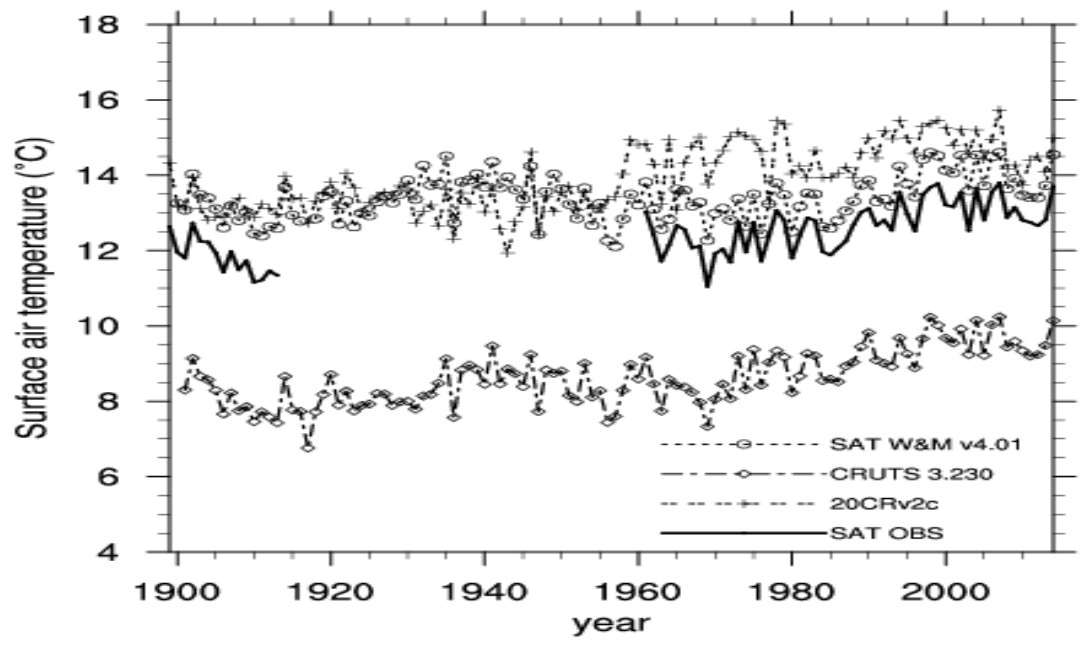


**Figure 5**. Annual mean SAT time series in Qingdao from 1899 to 2013 (Black solid line: SAT OBS; dashed line and
cross: 20CR v2c; dashed line and circles: SAT W&M v4.01; dashed line and rhombus: CRUTS3.230).

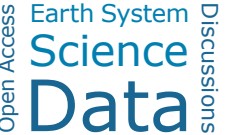

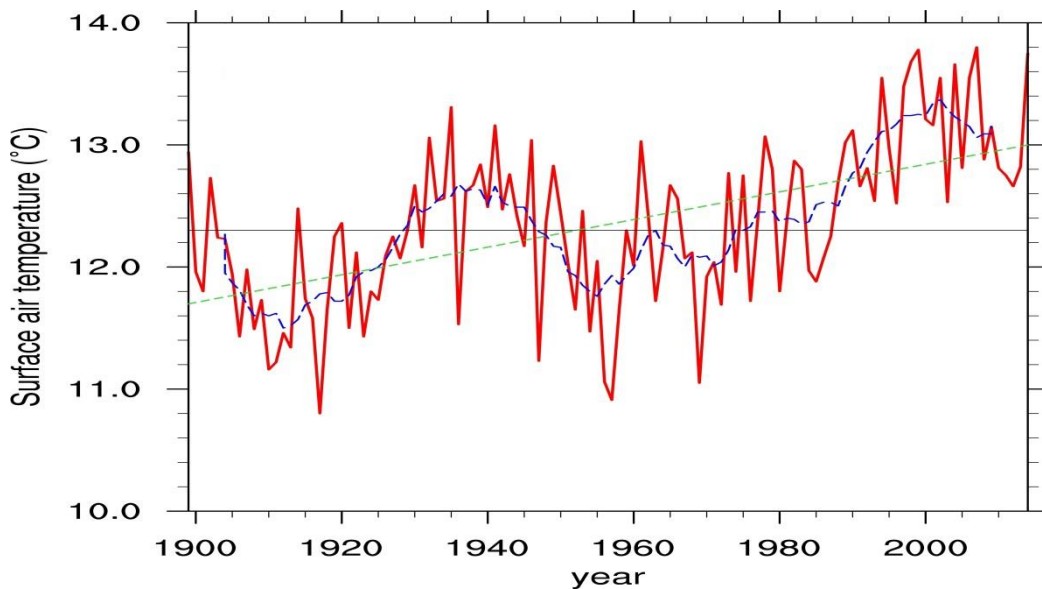


**Figure 6**. Construction of annual mean SAT ( ℃) in Qingdao for the period 1899-2014 (red solid line) and the 10 year
running mean (long dashed line). The short dashed line represents the long-term linear trend.   The thin black solid line
shows the climate annual mean SAT relative to 1961-1990.

**3.3 The temporal variations of SAT trend of the construction series**

235       The newly construction of annual mean SAT in Qingdao from 1899 to 2014 (Figure 6) exhibits a

warming rate in Qingdao over the last 116 years of   0.11  ℃ (10 yr) $^{-1}$, slightly larger than the global
mean warming rate over 1901-2012 (about 0.09  ℃ (10 yr) $^{-1}$, IPCC AR5, 2013). It indicates that the
warming trend is much significant (significant at 95% confidence intervals tested by the Mann-Kendall
test, p_value=1.1343e-09, which is less than 0.05).   From the time series, we also find that the coldest
years occurred in 1917 and 1969, with 11.03  ℃ and 11.05  ℃.   The warmest years occurred in 2007
and 1999, with annual means of 13.80  ℃ and 13.78  ℃.   Recently, Cao et al. (2013) published the first
homogenized set of SAT series during 1909-2010 of 16 stations in China. The calculated average
warming trend over the last hundred years based on this set of series is about 0.15  ℃ (10 yr) $^{-1}$, which is
consistent with the above result of Qingdao.

245        Since 2000, the SAT undergoes a decreasing trend, with the rate of -0.4 ℃ (10 yr) $^{-1}$, even if the

other SAT series (Figure 5) exhibit positive SAT anomalies for continuous 15 years. This slowdown or
cooling trend, or warming hiatus in this period are also found in regional and global scales (Morice et
al., 2012; Fyfe et al., 2013; Kosaka and Xie, 2013; Smith, 2013; Trenberth and Fasullo, 2013). These
studies further point out that stratospheric water vapor concentration, solar irradiance, Pacific Decadal
Oscillation, etc. may have led to this temporary variations.    If this local cooling tendency is related to
the "hiatus", or if may be related to regional and local conditions (e.g., increased aerosol presence) is
unknown and would need additional analysis.    It is also interesting to note that during 1899 to 1910
there is another decreasing trend, with the rate of -1.1 ℃ (10 yr) $^{-1}$, which is likely an expression of
natural variability, as we have no noteworthy global or regional forcing during that time. This cooling
rate is quite high, but the period extends only across 12 years, followed by a rapid warming for several
decades until about 1940 (Figure 6).

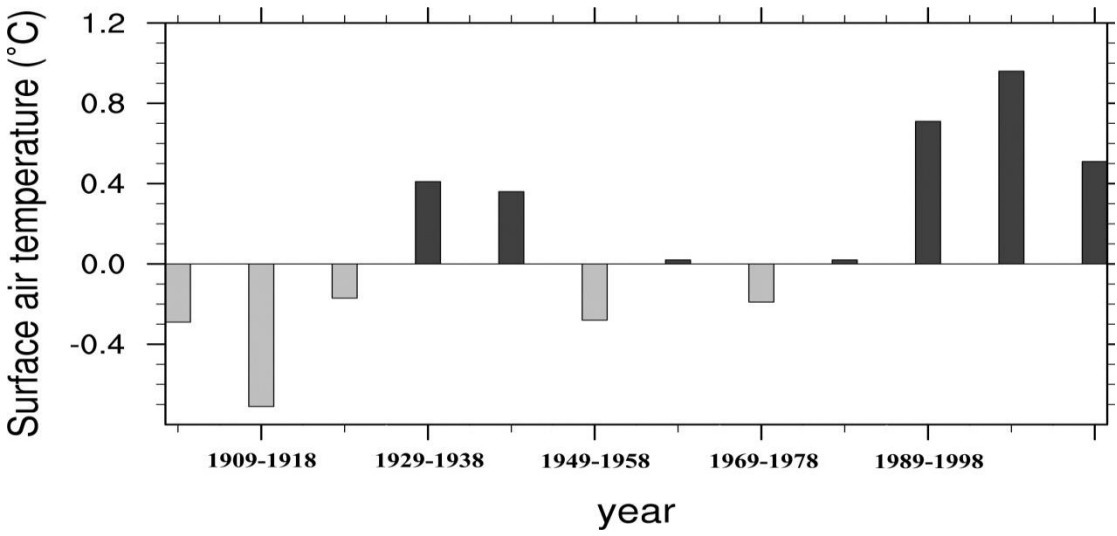


**Figure 7**. Average annual mean SAT anomalies for each decade ( ℃) in Qingdao during 1899-2014 relative to the
1961-1990 reference period (black line in Figure 6) (Gray bar means negative values and black bar means positive
values).




The constructed 10-year annual mean SAT from 1899-2014 are shown in Figure 7.  Five main
periods are associated with larger than normal SAT. The three maximum warm periods occurred at
1989-1998, 1999-2008 and 2009-2014. The average anomaly SAT of 1999-2008 is the largest which is
higher than normal for about 0.96 ℃. On the other hand, cold periods occurred in the following decades:
1899-1908, 1909-1918 and 1949-1958.    A wavelet analysis (figure omitted) suggests dominant and
persistent variations with time scales of about 40-80 years which is consistent with results of other studies
(Weng 2005, Wang and Zhang 2011).

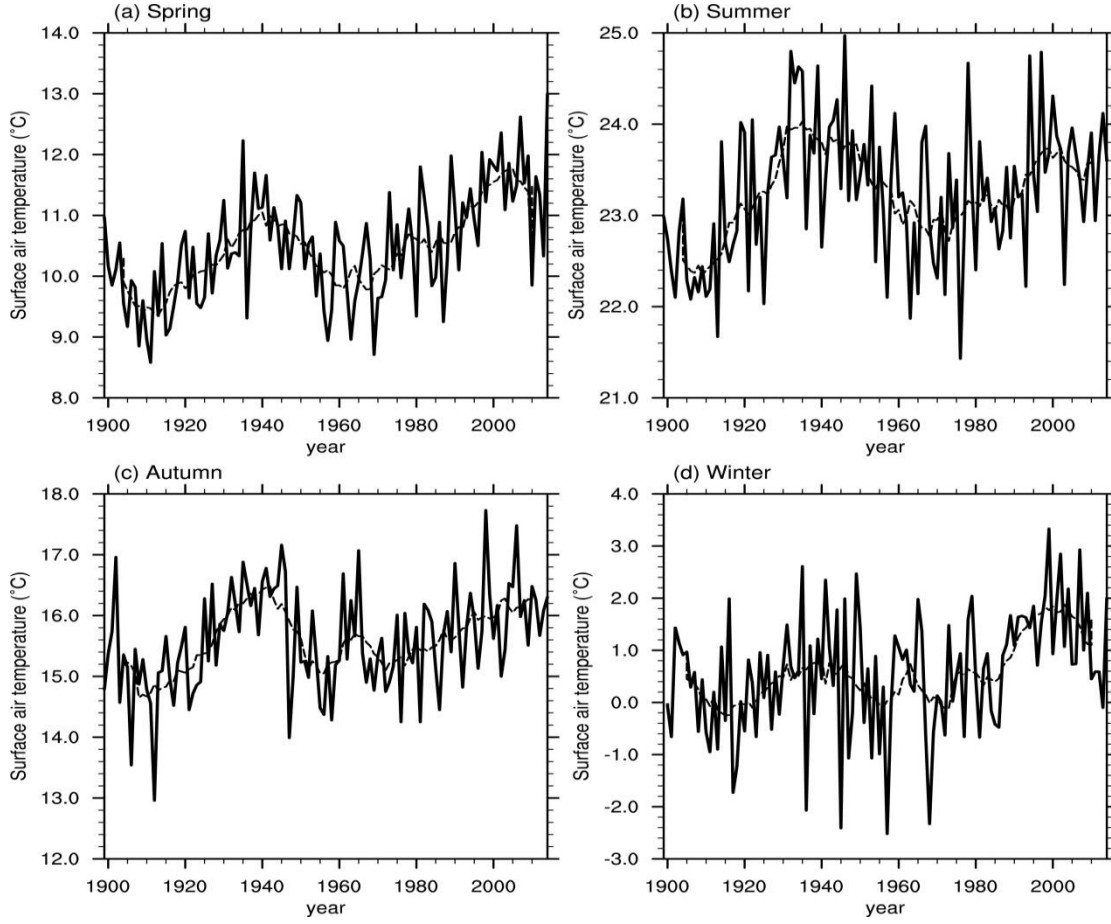






**Figure 8**. Construction of seasonal means SAT series ( ℃) in Qingdao for the period 1899-2014 (solid line) and the 10 year running mean (dashed line). (a) Spring, (b) Summer, (c) Autumn, (d) Winter.

The seasonal mean SAT time series are also shown in Figure 8, with the warming rate of about 0.13 ℃ (10 yr) $^{-1}$ in Spring, 0.05 ℃ (10 yr) $^{-1}$ in Summer, 0.07 ℃ (10 yr) $^{-1}$ in Autumn and 0.10 ℃ (10 yr) $^{-1}$ in Winter, respectively. These values suggest that the most significant warming occurred in Spring, followed by Winter. The interdecadal variations of the seasonal mean temperature in Spring and Winter appeared as a series of waves with a time scale of about 50-60 years. However, the trends in Summer and Autumn are much lower. The seasonal difference of the warming rate was also exhibited in other researches on SAT in other stations of China, or SAT in the whole of the China (Feng et al., 2009)

## 4. Conclusion and Discussion

Construction of a long-term homogeneous meteorological time series is essential for research into climate change. Using quality control, interpolation and homogeneity methods, we objectively establish a set of homogenized monthly mean SAT series in Qingdao of China from 1899 to 2014. Three data sets combined in this study, including the newly digitized observations of Qingdao station from German National Meteorological Service from 1899-1913, adjusted SAT W&M v4.01 from Delaware University during 1914 to 1959 and homogenized SAT data set from CMA during 1960 to 2014.

Based on the monthly SAT data, long-term changes in Qingdao of China are analyzed for the 1899-2014. Main conclusions are as follows: 1) The SAT in Qingdao has a significant warming trend of 0.11 ℃ (10 yr) $^{-1}$ during 1899-2014. 2) There are two periods with cooling trends, that is, 1899-1910 (-1.1 ℃ (10 yr) $^{-1}$) and 2000-2013 (-0.4 ℃ (10 yr) $^{-1}$). 3) There are seasonal differences of the warming rate with the largest warming rate in spring. These characters of the SAT variabilities in Qingdao agree well with the variabilities of SAT in the same region of China in the previous works.



In this study we only develop a set of homogenized monthly SAT in Qingdao.    Further efforts
should include:
1) A long-term series of other environment elements' observations in Qingdao and other stations in
China.
2) Acquisition of additional quality controlled data and metadata in the first half of the 20$^{th}$ century
and earlier for conditions in China by incorporate investigations and researches.
Based on such additional data, we can make further progress in our understanding of past, present,
and potential future climate change in the region. This will be addressed in future work.
From this study, we also have noticed that reconstruction and digitization of historical weather
observations is important for prolonging time series or fill gaps and improving the gridded or reanalysis
data set.    Furthermore, it is essential to be aware that metadata is important for homogenization of the
time series, especially in the earlier times without reference series. We therefore agree with Allan et al.
(2011 and 2016) that longer and more spatially and temporally-complete historical weather record could
be recovered, imaged and digitized to expand the observational database. There is still a long way to go.
A by-product of our work is that we have to confirm that data sets like 20CR should be examined
with care before using it for describing past variations. Certainly, a project like 20CR deserves are all
recognition, but in its present state the constructed data set still suffers from inhomogeneities prior to
1950, which hopefully will be overcome in future data sets constructed as global re-analysis for the
entire 20$^{th}$ century.

**Acknowledgments** This work was conducted by the lead author during a stay as visiting scientist at the
German Meteorological Service (Deutscher Wetterdienst, DWD) and Federal Maritime and



Hydrographic Agency (BSH) in German. We thank Hamburg University's Cluster of Excellence
CliSAP (Integrated Climate System Analysis and Prediction) for funding the stay.

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
