# Peer review of "Construction of surface air temperature series of Qingdao in China for the period 1899 to 2014"

_Earth System Science Data, 2017_

## Referee Comment (RC1) · Anonymous Referee #1 · 22 Nov 2017

Review ESSD-2017-58

Surface Air Temperature Qingdao

Important and positive to see these types of data sets emerge, both for the location and the century-long time scales.

Easy access to a very clean .txt file. Easy to reproduce Figure 6. A .csv file might represent a more familiar format to many users?

For comparison and reference purposes we should have the relevant WMO station number? If not Qingdao then nearest reliable station? Or perhaps CMA has its own station numbers and perhaps reference stations? If so, helpful to know how this data fits within the CMA system?

The descriptive manuscript seems to lack information on validation and uncertainties? For validation I looked at CRUTEM4 (for access, start from https://doi.org/10.5194/essd-6-61-2014). I plotted the CRUTEM4 gridbox anomaly values for Qingdao (derived from multiple stations within that grid box) with these ESSD-2017-55 data (see below).

[Figure]

One sees, not surprisingly, remarkable good correspondence, no doubt because for most time periods CRUTEM4 and this data set access identical source data. But the time period 1920 to 1940 suggests some discrepancies. Either CRUTEM4 seems low or Willmott and Matsuura seem high? Do the authors have an explanation? Do the authors have sufficient confidence in the quality of this data to suggest a revision to CRUTEM4? Could this particular station differ substantially from the grid-box average over those two decades?

In this data set we find, monthly and annually, precise temperature values with no hints of uncertainty. But uncertainties must have arisen in, at minimum: a) the original measurements at all time periods; b) the digitisation process (presumably involving optical character recognition) of the German charts; c) the Willmott and Matsuura interpolation and gridding processes; d) the more recent (and, one presumes, more accurate) CMA data processing; and e) the quality control and homogenisation processes described here. The authors should at least compile that information from the cited references as a guide and perhaps a caution to users of this time series? One would like to see error bars or coloured uncertainty ranges (presumably decreasing with time) on, for example, Figure 6. In my re-plot of data for Figure 6 I included a (not very high) correlation coefficient. The authors could do likewise and explain for readers and users the

various reasons (natural variability plus measurement uncertainty) for the values they expect and achieve.

One could exploit this data for additional information?  Perhaps a full analysis belongs in a separate science paper but, because of thrice daily data from 1899 to 1905 and hourly data from 1905 to 1914, the authors could at least hint or promote the possibilities of comparing early 20th century with present day daily temperature ranges.  They could also look at differential warming, nocturnal vs diurnal.  The authors mention rapid industrialisation in and around Qingdao.  Do the sub-daily data then and now reflect those changes?

---

## Referee Comment (RC2) · Anonymous Referee #2 · 27 Nov 2017

This paper is an important contribution to filling a data gap by means of newly digitized temperatures from Qingdao, China by means of data from colonial times in the late 19th century and up to World War I. The time series of Qingdao is of specific interest because there are 10 years of hourly observations from 1905 to 1914. This makes it possible to compare the diurnal cycle of temperature with present-day observations.

The paper is well-written, straightforward to understand and on a solid mathematical foundation. As such it deserves publication in Earth System Science Data. In terms of content, the only thing that is lacking is an estimate of uncertainties. How reliable are the early parts of the period? Does the high temporal resolution in 1905 to 1914 help to constrain uncertainties with respect to the other periods to to present-day climate? How much does the now dataset improve existing datasets, e.g. from CRUTEM?

I recommend that the paper should be published in ESSD, provided that the authors have addressed to remarks and comments outlined below.

Specific comments:

Line 85 (Table 1): Are the given times local times within the time zone or true local times (i.e. additionally take the longitude of the station into account? Given that Qingdao was a German colony, it appears straightforward to assume that the "Anleitung zur Anstellung und Berechnung meteorologischer Beobachtungen" (van Bebber, 1904) or a similar official publication was used as reference. These manuals state that hourly observations must be taken on the hour, but three times daily ("climate") observations must be taken at true local time, which depends on longitude. Given the fact that there are several different "climate" observation times (7-14-21, 8-14-22, etc.) and given the fact that true local time may differ almost one hour from zone time (if the station is on the western edge of the reference longitude "belt"), it is important to know when these observations were taken. To my knowledge, Beijing local time was used in all of China prior to 1913, but it appears plausible that the colony rather followed what today is UTC+8 (in which case the time difference would be negligible, since Qingdao is almost on 120 E). So, if at all possible, the authors should constrain the actual time used in these observations.

Lines 150/151: These are heights above sea level, I assume? Is any information available where the thermometers were situated with respect to the ground?

Lines 248/249: Wouldn't the most straightforward explanation for the "hiatus" be the increasing amounts of aerosols in the atmosphere? So I suggest to reformulate this paragraph.

Lines 267-269: Discussing wavelets with time scales of 80 years in a 100 year time series is rather close to overinterpretation.

There are a number of typos I would like the authors to correct. Also, there are a

number of less than optimal formulations. For example:

Line 18: Excel

Lines 18-21: Something is missing in this sentence.

Line 60: Delete "there are".

Line 61: Add question mark after the question.

Line 68: "...the archive of the..."

Line 138: "...are not efficient..."

Lines 310-314: I am aware of the restrictions of 20CR, but is one newly constructed time series really enough to state that there is a problem with 20CR?

---

## Author Comment (AC1) · 5 Feb 2018

Review ESSD-2017-58

Surface Air Temperature Qingdao

Important and positive to see these types of data sets emerge, both for the location and the century-long time scales.

\*\*\*\*\*\*\*\*\*\*\*\*\*\*\*\*\*\*\*\*\*\*\*\*\*\*\*\*\*\*\*\*\*\*\*\*\*\*\*\*\*\*\*\*\*\*\*\*\*\*\*\*\*\*\*\*\*\*\*\*\*\*\*\*\*\*\*\*\*\*\*\*\*\*

Reply to referees:

We appreciate the comments. And thanks a lot to the reviewer's works which have greatly improved our manuscript. The followed text is our point-by-point reply. The words with **blue colour are comments from referees**. The words with **black colour are the author's response.**

1. Easy access to a very clean .txt file. Easy to reproduce Figure 6. A .csv file might represent a more familiar format to many users?

    Thank you for your suggestion. DWD hosts the station data of Qingdao on https://dx.doi.org/10.5676/DWD/Qing_v1. It is policy of DWD to provide station data only in ASCII format. That's why we cannot provide the time series of Qingdao in the formarts *.csv and *.xls.

2. For comparison and reference purposes we should have the relevant WMO station number? If not Qingdao then nearest reliable station? Or perhaps CMA has its own station numbers and perhaps reference stations? If so, helpful to know how this data fits within the CMA system?

    The relevant WMO station number of Qingdao is 54857. The Qingdao station is a national meteorological reference and basic station which is located at $36^o 04´$ N, $120^o 20´$ E. And the station is 76 meters above the sea level.

3. The descriptive manuscript seems to lack information on validation and

uncertainties? For validation I looked at CRUTEM4 (for access, start from https://doi.org/10.5194/essd-6-61-2014). I plotted the CRUTEM4 gridbox anomaly values for Qingdao (derived from multiple stations within that grid box) with these ESSD-2017-55 data (see below).

[Figure]

One sees, not surprisingly, remarkably good correspondence, no doubt because for most time periods CRUTEM4 and this data set access identical source data. But the time period 1920 to 1940 suggests some discrepancies. Either CRUTEM4 seems low or Willmott and Matsuura seem high? Do the authors have an explanation? Do the authors have sufficient confidence in the quality of this data to suggest a revision to CRUTEM4? Could this particular station differ substantially from the grid-box average over those two decades?

Due to wars, station relocation, change of observation methods and instruments, as well as the sparse of the stations, there are great uncertainties in the temperature change before 1950s. Previous study point out that the observation SAT from 1914 to 1960 was discontinuous with many missing data (Cao et al., 2013). For Qingdao station, before 1960 the missing times of records were in July 1914 to March 1915; Sep 1937 to Jan 1938; and Jan 1951 to Dec 1960. That is one reason why the period of 1920 to 1940 has some discrepancies in the two dataset, CRUTEM4 and Willmott

and Matsuura. Annual mean SAT time series from Willmott and Matsuura and observations in Qingdao during 1916 to 1950 are shown in Figure 1. These originally observed SAT from 1916 to 1950 are obtained from China Meteorological Administration (CMA). Daily mean SAT in this period is defined by calculating the average of daily maximum and minimum temperature. Then the monthly or yearly mean SAT is calculated from the average of each daily mean temperature in a month or a year. All data from 1916 to 1950 has not been homogenized. It can be seen that the annual mean SAT series from Willmott and Matsuura has the similar climate variability with that of the observations. But the values of Willmott and Matsuura are slightly lower than the observations. Thus, the values of CRUTEM4 are much lower than observations. The conclusion agrees with the previous study (Li et al., 2016). Though CRUTEM4 update set out to reduce a known bias in the dataset, identified in multiple studies, by improving land coverage which is a good SAT product for us to analyze the global warming, for the climate change study in a city, CRUTEM4 may have more uncertainties due to the lower spatial resolution, only 5 degree grid.

[Figure]

Figure 1. Annual mean SAT time series in Qingdao from 1916 to 1950

4. In this data set we find, monthly and annually, precise temperature values with no hints of uncertainty. But uncertainties must have arisen in, at minimum: a) the original measurements at all time periods; b) the digitisation process (presumably involving optical character recognition) of the German charts; c) the Willmott and Matsuura interpolation and gridding processes; d) the more recent (and, one

presumes, more accurate) CMA data processing; and e) the quality control and homogenisation processes described here. The authors should at least compile that information from the cited references as a guide and perhaps a caution to users of this time series? One would like to see error bars or coloured uncertainty ranges (presumably decreasing with time) on, for example, Figure 6. In my re-plot of data for Figure 6 I included a (not very high) correlation coefficient. The authors could do likewise and explain for readers and users the various reasons (natural variability plus measurement uncertainty) for the values they expect and achieve.

Thank you very much for these very good recommendations. These investigations are from our point of view are expensive and would give the paper second focus. We would like to study this in a separate paper together with your next point.

5. One could exploit this data for additional information? Perhaps a full analysis belongs in a separate science paper but, because of thrice daily data from 1899 to 1905 and hourly data from 1905 to 1914, the authors could at least hint or promote the possibilities of comparing early 20th century with present day daily temperature ranges. They could also look at differential warming, nocturnal vs diurnal. The authors mention rapid industrialization in and around Qingdao. Do the sub-daily data then and now reflect those changes?

Thank you for your wonderful suggestion. Hourly observations in the earlier of 20$^{th}$ century make it possible to compare the extreme temperature and diurnal cycle of temperature with present-day observations. Here we finally chose the period from 1th Jan 1907 to 31th Dec 1913 with little missing data. Then we compare the maximum temperature (TX), minimum temperature (TN) and diurnal temperature range (Table 1) in the period from 1th Jan 1907 to 31th Dec 1914 to these in the period from 1th Jan 2007 to 31th Dec 2013 (100 hundred years interval). Hourly data from 1th Jan 2007 to 31th Dec 2014 are provided by the National Meteorological Information Center of the China Meteorological Administration (CMA). Yearly mean daily TX (TN/DTR) is defined by calculating the average of each daily TX (TN/DTR) in a year. Then the differences of TX, TN, DTR between the two periods are shown in Figure 2. In Figure

2, the TX and TN are found to have significantly increased at the range of 1.2~2.7℃ and 1.0~2.4℃. It means that relative to the years at the beginning of the 20th century, both of the TX and TN rise by ~2.0℃ at the beginning of the 21th century. However, there is no notable increase or decrease in the DTR. Global annually averaged surface air temperature has increased by 1.0 over the last 115 years (1901-2016) according to the Climate Science Special Report 2017 of USA. Compared to the global warming, extreme temperature warming in Qingdao is much stronger which may be caused by the rapid industrialization in and around Qingdao.

Table 1 Definitions of temperature indices used in this study.

| Index | Descriptive name | Definition | Units |
|-------|-----------------|------------|-------|
| TX | maximum temperature | Yearly mean daily maximum temperature | ℃ |
| TN | minimum temperature | Yearly mean daily minimum temperature | ℃ |
| DTR | Diurnal temperature range | Yearly mean difference between daily maximum and minimum | ℃ |

[Figure]

Figure 2 Differences of TX, TN, DTR between the period of 1907-1913 and the period of 2007-2013

---

## Author Comment (AC3) · 5 Feb 2018

Reply to referees:

   We appreciate the comments. And thanks a lot to the referee's works which have greatly improved our manuscript. The followed text is our point-by-point reply. The words with blue colour are **comments from referee**. The words with **black colour are the author's response.**

Specific comments:

1. Line 85 (Table 1): Are the given times local times within the time zone or true local times (i.e. additionally take the longitude of the station into account? Given that Qingdao was a German colony, it appears straightforward to assume that the "Anleitung zur Anstellung und Berechnung meteorologischer Beobachtungen" (van Bebber, 1904) or a similar official publication was used as reference. These manuals state that hourly observations must be taken on the hour, but three times daily ("climate") observations must be taken at true local time, which depends on longitude. Given the fact that there are several different "climate" observation times (7-14-21, 8-14-22, etc.) and given the fact that true local time may differ almost one hour from zone time (if the station is on the western edge of the reference longitude "belt"), it is important to know when these observations were taken. To my knowledge, Beijing local time was used in all of China prior to 1913, but it appears plausible that the colony rather followed what today is UTC+8 (in which case the time difference would be negligible, since Qingdao is almost on 120 E). So, if at all possible, the authors should constrain the actual time used in these observations.

   Thank you for your careful review. Unfortunately we can find in the original hand written observation books as well as in "Deutsche Ueberseeische Beobachhtungen"

(German observations in overseas) only "7a, 2p, 7p" and no information on whether it is "true local time" or "time zone". The hourly data are taken from self-recording instruments and we think that a daily mean of 24 values should be always the same, independent from the minutes before or after the full hour. In order to find out what the local time mean, we look through several historical handwritten journals. Results show that at the end of 19 century, China's coastal ports began to use Greenwich Standard Time, that is, the local time of $120^{o}$ E as the standard time, and called "coast time". The coast time facilitated navigation and trade in colony period of China. At present, China uses Beijing time, that is, the time zone of the East eight area as the standard time. Qingdao is almost on $120^{o}$ E. Thus, the time difference would be negligible between the colony period and present.

2. Lines 150/151: These are heights above sea level, I assume? Is any information available where the thermometers were situated with respect to the ground?

[Figure]

Abb. 1.1: Historische Karte von der Umgebung von Tsingtau im Jahr 1912

[Figure]

**Abb. 1.2:** Geographische Position der verschiedenen Standorte der Klimastation laut der damaligen ermittelten geographischen Koordinaten, Karte: Google Maps

In the earlier time from Jan 1899 to Apr 1905, SAT was observed at 36 °04 ΄, 120 °17 ΄, which was a harbor near the coast of Qingdao (Figure 2). From May 1905 to Mar 1914, SAT was observed at 36 °04 ΄, 120 °19 ΄, which was a small hill. The meteorology station was located at the top of this hill, without the effect of buildings and forest (Figure 1 and 2)

3. Lines 248/249: Wouldn't the most straightforward explanation for the "hiatus" be the increasing amounts of aerosols in the atmosphere? So I suggest to reformulate this paragraph.

The sentences in L248-249 are modified as follows in the revised paper:
The Intergovernmental Panel on Climate Change Fifth Assessment Report (IPCC AR5) pointed out that during the recent years (1998-2013), the global warming rate has slowed down. This period has been discussed in a number of papers (Kosaka and Xie; Smith) and has known as "hiatus" (Fyfe et al.) or "global slowdown" (Guemas et al). Recent studies indicate that the recent "warming hiatus" is also found in Chinese SAT changes (J Wang et al., 2014; Li Q X et al., 2015). Proposed mechanisms for the

recent warming hiatus at the global scale are still under debate, including changes in deep-ocean heat uptake, anthropogenic aerosols, solar variations, volcanoes, and sea surface temperature (Meehl et al., 2011; Kaufamann et al., 2011; Kosaka and Xie, 2013). At regional and local scales, other factors may also play a significant role, which makes attribution challenging.

4. Lines 267-269: Discussing wavelets with time scales of 80 years in a 100 year time series is rather close to overinterpretation.

We delete the sentence "A wavelet analysis (figure omitted) suggests dominant and persistent variations with time scales of about 40-80 years which is consistent with results of other studies (Weng 2005, Wang and Zhang 2011)."

5. There are a number of typos I would like the authors to correct. Also, there are a number of less than optimal formulations. For example: Line 18: Excel

We modify "excel" into "Excel" in the revision.

6. Lines 18-21: Something is missing in this sentence.

The data is provided and archived by Deutscher Wetterdienst (DWD) web page under overseas stations of the Deutsche Seewarte (http://www.dwd.de/EN/ourservices/overseas_stations/ueberseedoku/doi_qingdao.html) in the form of ASCII. Users also can freely obtain the data at https://dx.doi.org/10.5676/DWD/Qing_v1.

7. Line 60: Delete "there are".

We delete the two words "there are"

8. Line 61: Add question mark after the question.

We add the question mark at the end of each question.

We modify "archive" into "the archive of the".

We modify "is not efficient" into "are not efficient".

11. Lines 310-314: I am aware of the restrictions of 20CR, but is one newly
    constructed time series really enough to state that there is a problem with 20CR?

Thank you for your reminding.   In our work, the monthly mean SAT data of the 20CR with a horizontal resolution of 2 °in both longitude and latitude are interpolated to the locations of meteorological stations to compare with the newly constructed long-term SAT series. To be honest, there are some uncertainties in the processes of interpolation and gridding, considering that the horizontal resolution is much low. We agree with you that just one newly constructed time series really enough to state that there is a problem with 20CR. So we delete this paragraph in the revision paper.